# Transcriptome Analysis of Maize Ear Leaves Treated with Long-Term Straw Return plus Nitrogen Fertilizer under the Wheat–Maize Rotation System

**DOI:** 10.3390/plants12223868

**Published:** 2023-11-16

**Authors:** Jun Li, Jintao Liu, Kaili Zhu, Shutang Liu

**Affiliations:** 1College of Agronomy, Qingdao Agricultural University, Qingdao 266109, China; 17863940373@163.com; 2School of Engineering, Universidad de Almería, ES04120 Almería, Spain; jintaol@163.com; 3College of Resources and Environment, Qingdao Agricultural University, Qingdao 266109, China

**Keywords:** maize, straw return, nitrogen fertilizer, transcriptome

## Abstract

Straw return (SR) plus nitrogen (N) fertilizer has become a practical field management mode to improve soil fertility and crop yield in North China. This study aims to explore the relationship among organic waste, mineral nutrient utilization, and crop yield under SRN mode. The fertilizer treatments included unfertilized (CK), SR (straws from wheat and corn), N fertilizer (N), and SR plus N fertilizer (SRN). SRN treatment not only significantly increased the grain yield, net photosynthetic rate, and transpiration rate but also enhanced the contents of chlorophyll, soluble sugar, and soluble protein and increased the activities of antioxidant enzymes but reduced intercellular CO_2_ concentration and malondialdehyde (MDA) content when compared to other treatments. There were 2572, 1258, and 3395 differentially expressed genes (DEGs) identified from the paired comparisons of SRvsCK, NvsCK, and SRNvsCK, respectively. The transcript levels of many promising genes involved in the transport and assimilation of potassium, phosphate, and nitrogen, as well as the metabolisms of sugar, lipid, and protein, were down-regulated by straw returning under N treatment. SRN treatment maintained the maximum maize grain yield by regulating a series of genes’ expressions to reduce nutrient shortage stress and to enhance the photosynthesis of ear leaves at the maize grain filling stage. This study would deepen the understanding of complex molecular mechanisms among organic waste, mineral nutrient utilization, crop yield, and quality.

## 1. Introduction

Massive amounts of chemical nitrogen (N) fertilizers have been applied to agricultural soil for the past forty years. Chemical N plays an important role in enhancing agricultural food production [1]; however, low N utilization efficiency (NUE), nitrate leaching, and severe environmental issues such as the emission of nitrous oxide and ammonia into the atmosphere occurred due to excessive application of chemical N fertilizer [2]. Therefore, it is necessary to search for alternative ways to reduce the overuse of synthetic fertilizers. 

The winter wheat-summer maize rotation system is the main cropping mode in North China, producing 518 million tons of straw annually [3,4]. In the past, more than 30% of crop residues were frequently regarded as organic wastes, which were simply burned in fields after harvest and resulted in serious air pollution [5,6]. Nowadays, its beneficial utilization has received much attention because the rotted straw can provide crops with suitable nutrients and various salinities [7]. Straw return (SR) to the field can not only increase soil organic carbon (SOC), maintain soil fertility, and promote crop yield [8,9] but also improve the defense response of plants to diseases [10]. 

SR plus N fertilizer (SRN) practice has a significant influence on soil fertility and crop yield. In a seven-year-old field experiment, Chen et al. [11] reported that continuous SRN treatment improved maize yield by 154% and meanwhile dramatically increased soil nutrients, including total soil N, available P, available K, enzyme activities (such as invertase, urease, and cellulose) and organic carbon. Yang et al. [8] also demonstrated that SR significantly reduced soil N leaching at 30–90 cm soil depth, increased the NUE, and thus improved the yields of rice and wheat.

Maize (*Zea mays* L.) is one of the important food, feed, and industrial crops in the world. Although the positive impacts of SRN on soil fertility and crop yield were well documented, little is known regarding the molecular relationship among organic waste, mineral nutrient utilization, and crop yield under SRN mode in long-term field conditions. Till now, no transcriptome analysis of maize responsive to SRN treatment under Wheat–Maize rotation system has been reported in field conditions so far. Considering that ear leaves contribute a substantial proportion of photosynthetic products to maize kernels [12], we conducted transcriptome analysis of ear leaves under different fertilizer treatments using RNA-Seq techniques. The study would provide insights into the molecular mechanism of the enhancements of mineral nutrient utilization and crop yield via SR and its combination with N fertilizer. Also, the study revealed the candidate genes involved in mineral nutrient transport and assimilation and the metabolisms of sugar, lipid, and protein.

## 2. Results

### 2.1. Grain Yield and Photosynthetic Parameters under Various Fertilizer Treatments

Compared to the control, the grain yields under SR, N, and SRN treatments in two consecutive years were enhanced by 29%, 71%, 156.8%, and 11.3%, 58.5%, and 131%, respectively, in two consecutive years (Table 1). Long-term SRN treatment exhibited the highest maize grain yield among all treatments. Similarly, SRN treatment also showed higher chlorophyll content, net photosynthetic rate, transpiration rate, and total conductance to water, as well as lower intercellular CO_2_ concentration and total conductance to CO_2_ when compared to other treatments (Appendix A). Compared to the control, the chlorophyll content, net photosynthetic rate, transpiration rate, and total conductance to water under SRN treatment were significantly enhanced by 106%, 52%, 58%, and 45%, respectively, whereas the intercellular CO_2_ concentration and total conductance to CO_2_ were decreased by 7.5% and 14%, respectively. 

### 2.2. Physiological Changes of Ear Leaves Responsive to Various Fertilizer Treatments

As shown in Figure 1a,b, the contents of soluble sugar and soluble protein were quite lower for maize plants under CK than under fertilizer treatments but reached the maximum under SRN treatment. For instance, the contents of soluble sugar and soluble protein under SRN treatment were 7.3 times and 2.4 times the control, respectively. Besides, maize plants under CK treatment exhibited higher MDA content and lower CAT and POD activities than those under the other three fertilizer treatments, and the plants under SRN showed the lowest MDA content and the highest CAT and POD activities among three treatments (Figure 1c–e). MDA contents under SR, N, and SRN treatments were respectively decreased by 46%, 42%, and 65%, respectively, while CAT and POD enzyme activities were respectively enhanced by 314%, 222%, 571%, 42%, 35%, and 76% when compared to the control. 

### 2.3. Transcriptome Sequencing 

After removing the low-quality reads and adaptor sequences, we obtained a total of 974,889,944 clean reads. All clean reads contained 142.0922 Gb with a GC content of above 49%, a Q20 level of above 97.21%, and a Q30 level of above 92.96% (Appendix A). For reads mapping, approximately 75% (132,812,884/176,794,952) of the reads were mapped to the maize B73 genome, and about ~65% of the mapped reads (86,264,550/132,812,884) were uniquely mapped (Appendix A). 

### 2.4. Identification of DEGs Responsive to Various Fertilizer Treatments 

The clean reads from each of the three fertilizer treatments were all compared to those from the control, that is, SR versus CK (SRvsCK), N versus CK (NvsCK), SRN versus CK (SRNvsCK), and pairwise comparison was performed. The genes meeting *p*-value < 0.05 and ∣log_2_FC∣ ≥ 1.0 were chosen as DEGs for further analysis (Appendix A). Finally, we identified 2572 DEGs (1386 up-regulated and 1186 down-regulated) from SRvsCK, 1258 DEGs (704 up-regulated and 554 down-regulated) from NvsCK, and 3395 DEGs (1817 up-regulated and 1578 down-regulated) from SRNvsCK (Figure 2a). There were 223 common DEGs for the three groups of paired comparisons (Figure 2b). The correlation coefficient between RNA-Seq data and qRT-PCR data was 0.93, indicating that the transcriptome sequencing results are meaningful (Figure 2c). 

### 2.5. GO and KEGG Analysis of the DEGs Responsive to Fertilizer Treatments 

GO enrichment analysis of the DEGs demonstrated that the detected GO terms (*p*-value < 0.05) were involved in three categories: biological process (BP), cellular component (CC), and molecular functions (MF) (Appendix A). Many GO terms like “cell communication” and “response to abiotic stimulus” in biological processes, “chloroplast part” and “mitochondrion” in cellular components, “zinc ion binding” and “protein serine/threonine kinase activity” in molecular functions from SRNvsCK and SRvsCK were exactly the same. In biological processes, some special GO terms were found in one comparison group, like “lipid biosynthetic process” from SRNvsCK, “cellular amino acid metabolic process”, and “organic acid biosynthetic process” from NvsCK and “carbohydrate biosynthetic process” and “response to nutrient levels” from SRvsCK. The numbers of KEGG pathways (*p*-value < 0.05) detected from SRvsCK, NvsCK, and SRNvsCK are 15, 12, and 15, respectively (Appendix A). A few of these KEGG pathways, like “starch and sucrose metabolism” and “nitrogen metabolism”, are commonly in three groups of paired comparisons, but the others are especially found in one or two comparison groups. For example, “carotenoid biosynthesis” and “sphingolipid metabolism” were from SRvsCK, “biosynthesis of amino acids” was from NvsCK, and “MAPK signaling pathway”, “plant hormone signal transduction”, and “fatty acid elongation” were special from SRNvsCK. 

### 2.6. DEGs Involved in Mineral Nutrient Transport and Assimilation

According to the criteria of *p*-value < 0.05 and ∣log_2_FC∣ ≥ 1.0, the numbers of DEGs involved in potassium (K) transport were 10 (2 up-regulated and 8 down-regulated), 1 (1 up-regulated), 11 (2 up-regulated and 9 down-regulated) from SRvsCK, NvsCK and SRNvsCK, respectively, the DEGs involved in phosphorus (P) transport were 3 (1 up-regulated and 2 down-regulated), 3 (3 up-regulated), 2 (1 up-regulated and 1 down-regulated) from SRvsCK, NvsCK and SRNvsCK, respectively, and the DEGs involved in nitrogen (N) transport and assimilation were 3 (3 up-regulated), 7 (7 up-regulated), and 2 (2 up-regulated) from SRvsCK, NvsCK and SRNvsCK, respectively (Figure 3a–c). For these DEGs with ∣log_2_FC∣ ≥ 2.0, there were four DEGS involved in K transport, one involved in P transport, and seven involved in N transport and assimilation (Table 2). *GRMZM2G477457* encoding HAK3 and *GRMZM2G425999* encoding HAK4 were found to be significantly down-regulated, while *GRMZM2G122584* encoding potassium/sodium hyperpolarization-activated cyclic nucleotide-gated channel 1 was significantly up-regulated in SR and SRN treatments when compared with the control. However, no changes were observed in the N treatment. *GRMZM2G154090* encoding Pht2 was the sole gene that was significantly up-regulated under SR, N, and SRN treatments when compared with the control. Among seven DEGs involved in N transport and assimilation, two were involved in N transport, three were involved in N reduction, and the remaining two were involved in nitrite reduction. *GRMZM2G455124* encoding NRT2.3 and *NR* encoding nitrate reductase were significantly up-regulated in SR, N, and SRN treatments when compared with the control. Additionally, another four genes are also found to be up-regulated in N treatment but not in SR and SRN treatments.

### 2.7. DEGs Involved in Sugar Metabolism, Fatty Acid Metabolism and Amino Acids Biosynthesis

Based on the threshold of *p*-value < 0.05 and ∣log_2_FC∣ ≥ 1.0, we found many DEGs related to sugar metabolism, fatty acid metabolism, and biosynthesis of amino acids. The numbers of DEGs involved in sugar metabolism were 19 (5 up-regulated and 14 down-regulated), 17 (11 up-regulated and 6 down-regulated), and 28 (11 up-regulated and 17 down-regulated) from SRvsCK, NvsCK, and SRNvsCK, respectively, DEGs involved in fatty acid metabolism were 7 (2 up-regulated and 5 down-regulated), 4 (2 up-regulated and 2 down-regulated), and 13 (3 up-regulated 10 down-regulated) from SRvsCK, NvsCK, and SRNvsCK, respectively, and those involved in the biosynthesis of amino acids were 13 (7 up-regulated and 6 down-regulated), 21 (18 up-regulated and 3 down-regulated), and 23 (18 up-regulated and 5 down-regulated), from SRvsCK, NvsCK and SRNvsCK, respectively (Figure 4a–c). Among the DEGs above, 25 met the threshold of ∣log_2_FC∣ ≥ 2.0. Fourteen of the 25 DEGs were related to sugar metabolism, three were related to fatty acid metabolism, and eight were related to the biosynthesis of amino acids (Table 3). The fourteen DEGs relevant to sugar metabolism were involved in four groups: starch metabolism, cellulose metabolism, trehalose metabolism, and sugar transport. Compared to the control, *AGPL2* encoding glucose-1-phosphate adenylyltransferase large subunit 2 was significantly up-regulated in response to three fertilizer treatments. Contrary to *AGPL2*, *GRMZM2G008263* encoding granule-bound starch synthase II, a precursor, *GRMZM2G086845* encoding fructokinase 1, and *GRMZM6G738249* encoding trehalose-phosphate phosphatase J were significantly down-regulated in SR and SRN treatments, but their expression level did not change in N treatment. Compared to the control, *GRMZM2G157675* encoding SWEET was the sole gene that was significantly down-regulated in N treatment. Except for *GRMZM2G157675*, two genes encoding beta-glucosidase 1 (*GRMZM2G016890* and *GBA1*), *GRMZM2G066162* encoding endoglucanase 12-like and *GRMZM2G063824* encoding carbohydrate transporter/sugar porter were significantly up-regulated in N treatment, but their expression level kept the same as in SR and SRN treatments. There were three DEGs (*GRMZM2G178014*, *GRMZM2G022558,* and *GRMZM2G160417*) involved in fatty acid metabolism. *GRMZM2G160417* has not been characterized so far. The other two genes were regulated oppositely in SR and SRN treatments: *GRMZM2G178014* encoding alpha/beta-Hydrolases superfamily protein was up-regulated while *GRMZM2G022558* encoding fatty acid elongase 1 was down-regulated. The expression trends of *GRMZM2G178014* and *GRMZM2G022558* were not observed in N treatment. Among the eight DEGs involved in amino acid biosynthesis, *GRMZM2G028535*, *GRMZM2G061777*, and *Rpi2* were significantly up-regulated in all the fertilizer treatments. Except for down-regulated *GRMZM2G153536*, the other three genes (*GRMZM2G053669*, *GRMZM2G078472,* and *GRMZM2G013430*) detected in N treatment were significantly up-regulated, but their expression level did not change in SR and SRN treatments. On the contrary, *GRMZM2G124963* encoding alanine aminotransferase 10 exhibited significant up-regulation in SR and SRN treatments but no difference in N treatment. 

## 3. Discussion

Straw return to soil has been a common agricultural practice in China [13]. In the present study, maize plants growing in unfertilized plots exhibited significantly reduced chlorophyll, soluble sugar, and soluble protein contents and lowered antioxidant enzyme activities but highly enhanced MDA content. It indicated that these plants were subjected to severe osmotic and oxidative stresses due to nutrient deficiency. However, this adverse growth status is largely alleviated by the addition of SR, N fertilizer, and SRN, of which SRN had the best effect. Maize plants under SRN treatment exhibited higher antioxidant activities and higher contents of chlorophylls and osmotic materials but lower MDA contents, and thus, the photosynthetic capacity and yields were improved. This further explains the reason why SRN is a common practice to improve soil fertility and crop yield in North China Plain. Additionally, this result agrees with our GO and KEGG analyses that are rich in “response to abiotic stimulus”, “starch and sucrose metabolism”, “nitrogen metabolism”, and so on. Su et al. [7] reported that rotten crop straw could provide various nutrients except for nitrogen for crops. It is conceivable that SR or SRN would induce the differential expression of more genes than sole N application. Our transcriptome analysis verified that the number of DEGs from the NvsCK comparison is much less than those from SRvsCK or SRNvsCK comparison group.

N, P, and K are essential nutrients for plant growth and crop yield. Their utilization efficiency in crops is largely dependent on their capacities of absorption and translocation in crops [14]. Here, only four genes related to K transport were found to be highly regulated by different fertilizer treatments, and three of them were significantly down-regulated. This is consistent with the previous study reported by Shen et al. [15], in which the transcript levels of most K transporters and channels were reduced to be responsive to low K treatment. In maize, there are 27 genes encoding high-affinity K transporter (HAK) [16]. Here, we found that *HAK3* and *HAK4* were significantly down-regulated in SR and SRN treatments but not in N treatment. *GRMZM2G122584* encoding potassium/sodium hyperpolarization-activated cyclic nucleotide-gated channel 1 was the sole gene that was found to be highly up-regulated in SR and SRN treatments but not in N treatment. This suggests that the supplement of straw returning induces more gene expression than sole N treatment. Further study is needed to clarify their roles in K homeostasis in response to low K stress. 

H_2_PO_4_^−^, a major phosphate (Pi) source, is taken up via Pi transporters in the plasma membrane of root cells. So far, five Pi transporter families have been identified [17]. Here, *Pht2* encoding a Pht2 protein was found to be significantly up-regulated in all treatments, but the magnitude in SR and SRN treatments was much lower than that in the N treatment. This suggests that *Pht2* might play crucial roles in the Pi uptake and translocation, while the function was repressed to some extent by the supplement of SR. 

Low N utilization efficiency (NUE) has become a major constraint in crop production worldwide [18]. NUE is largely determined by nitrate uptake and nitrate reduction [19]. The former is mediated by nitrate transporters (NRT), and the latter, nitrate, is firstly degraded into nitrite and then to ammonium, catalyzed by nitrate (NR) and nitrite reductase (NIR), respectively. There are two types of nitrate transporters: one is low-affinity NRT1s, and the other is high-affinity NRT2s [18]. In sorghum, the low-N tolerant genotype showed higher transcript levels of NRT2s than the low-N sensitive genotype [20]. Here, eight genes, including three genes encoding nitrate transporters, three genes encoding nitrate reductase enzymes, and two genes encoding nitrite reductase enzymes, were significantly up-regulated in response to various fertilizer treatments. Consistent with the previous results [20], *GRMZM2G455124* and *GRMZM5G878558* showed significant up-regulation expression levels in all treatments, indicating that they played important roles in nitrate uptake and assimilation to deal with nutrient deficiency stresses. In the present study, the number and magnitude of DEGs in NvsCK comparison were much higher than those in SRvsCK and SRNvsCK comparisons, indicating that the addition of straw inhibited the expression levels of some N-related genes. In addition, the down-regulation of DEG genes related to K transporters and the up-regulation of DEG genes related to N transporters and P transporters responsive to nutrient shortage confirmed the important role of N and P in promoting the synthesis of key cellular molecules such as nucleic acid, protein, and ATP and maintaining the basic life activities of cells [21,22].

At the maize grain-filling stage, the ear-leaf plays a major role in providing photosynthetic assimilates for grain yield. It is conceivable that the transcriptional levels of a large number of sugar metabolism-related genes varied in response to various fertilizer treatments. Our transcriptome data showed that a total of 46 DEGs were significantly regulated, which is in agreement with vigorous photosynthesis in leaves. The lower ratio of up- or down-regulated DEGs in SR and SRN treatments when compared to N treatment suggested that the addition of straw into the field inhibited the expression levels of some sugar metabolism-related genes. 

Starch is the main storage carbohydrate in higher plants. ADPG pyrophosphorylase (AGP) is the rate-limiting enzyme that catalyzes the generation of precursor molecules for starch biosynthesis-ADPG [23]. Here, *AGPL2* encoding AGP large subunits were found to be significantly up-regulated in all treatments, indicating that it played a positive role in the conversion of photosynthetic products into starch upon different fertilizer treatments. However, another gene (*GRMZM2G008263*) involved in starch biosynthesis was significantly down-regulated in SR and SRN treatments but not detected in N treatment, suggesting that the addition of straw inhibited its expression. Many starch-degrading enzymes, including starch phosphorylase, α- and β-amylases, are engaged in starch breakdown. Here, we found that *GRMZM2G025833* encoding β-amylase was significantly down-regulated in SRN treatment but not detected in N and SR treatments, suggesting that SRN treatment was beneficial to starch accumulation in maize leaves. This explains the reason in part why maize plants under SRN treatment exhibited higher kernel yields.

As the main component of plant cell walls, cellulose is one of the most widely distributed and abundant polysaccharides in nature. Dong et al. [24] reported that late sowing increased the lodging resistance of wheat plants by up-regulating the contents of lignin and cellulose, which is intriguing. Here, three genes (*GRMZM2G016890*, *GBA1,* and *GRMZM2G066162*) involved in cellulose metabolism were found to be significantly up-regulated in N treatment but not detected in SR and SRN treatments. Also, other genes, such as *GRMZM2G086845*, *GRMZM6G738249*, etc., were significantly down-regulated in SR and SRN treatments but not detected in N treatment. Trehalose is a kind of non-reducing disaccharide that functions as a stress protectant in many organisms, such as primitive plants. Trehalose synthesis is catalyzed by TPS and TPP via Tre6P [25]. Here, *GRMZM6G738249* encoding trehalose-phosphate phosphatase J and *GRMZM2G140078* encoding trehalose-6-phosphate phosphatase two were found to be significantly down-regulated in SR and SRN treatments, but no differential expression was detected in N treatment, indicating that trehalose biosynthesis is highly regulated by the addition of straw and (or) N fertilizer thus to cope with stresses caused by nutrient limitation. Sugar transport from leaves to sink tissues is crucial for yield formation. SWEET (Sugar Will Eventually be Exported Transporter) has been considered to be involved in the apoplastic loading of sugar [26]. Yang et al. [27] reported that *CoSWEET12* from *Camellia oleifera* was involved in the regulation of sugar transport. In this work, two sugar transporter genes (*GRMZM2G157675* and *GRMZM2G063824*) showed quite opposite expression patterns in N treatment, but their expression level did not change in SR and SRN treatment. It indicated that sugar transport in maize leaf was highly regulated and responsive to different fertilizer treatments at the maize grain-filling stage. 

Fatty acids (FA) participate in the construction of cell membrane systems, stress resistance, nutritional quality improvement, and so on. Here, we found that the relevant DEG number in the N treatment was quite less than those in SR and SRN treatments. This is in agreement with the result of a previous study, which showed that some nutrients other than N promote FA biosynthesis and accumulation [28]. In FA biosynthesis, FA elongase (FAE) catalyzed the germination of long-chain fatty acids. Shi et al. [29] reported that *FAE1* mutation led to changes in the composition of fatty acid in *Brassica napus*. Here, *GRMZM2G022558,* involved in the production of long-chain FA, exhibited significantly down-regulated expression levels in SR and SRN treatments, but no differential expression was observed in N treatment. Fang et al. [30] reported that *GRMZM2G022558* participated in the biosynthesis of long-chain fatty acid by regulating the ratio of C18:0/C20:0 in a direct way. Conversely, *GRMZM2G178014* related to the elongation of FA carbon chain were significantly up-regulated in SR and SRN treatments. 

In plants, amino acids are important nitrogen storage compounds [31]. Our transcriptome analysis demonstrated that a total of 39 genes involved in the biosynthesis of amino acids expressed differentially, suggesting that amino acid biosynthesis was much more active at the maize grain-filling stage. The DEG number, especially the up-regulated DEGs, was quite less in SR treatment than in other treatments, indicating that straw nutrients were not sufficient for crop growing and N fertilizer application is necessary. Proline can serve as an osmoprotectant to cope with stresses caused by drought, salt, etc. [32]. In the present study, two genes (*GRMZM2G028535* and *GRMZM2G061777*) involved in proline biosynthesis were significantly up-regulated in fertilizer treatments, and meanwhile, the GO and KEGG analysis revealed that many regulated genes were related to plant defense response in all treatments. Asparagine plays a major role in nitrogen recycling and transport in many organs, such as developing seeds and vegetative and senescence organs [33]. Two genes (*GRMZM2G053669* and *GRMZM2G078472*) involved in aspartate biosynthesis were found to be significantly up-regulated only in N treatment but not in the other two fertilizer treatments, suggesting that straw addition adversely affected these genes’ expression. Histidine has been considered to play a role in enhancing heavy metal tolerance in plants and bacteria [34,35]. Here, *Rpi2* involved in histidine biosynthesis is significantly up-regulated in all treatments, indicating that it might have participated in plant defense to cope with nutrient deficiency. 

## 4. Materials and Methods

### 4.1. Experiment Site, Crop Material and Design

The field experiment was carried out at a long-term fertilization experimental site located at Laiyang City, Shandong Province, China (36°10′ N, 120°31′ E). The experimental site was established in 2009 with the aim of evaluating the effect of long-term straw returning on soil quality and crop production. The initial nutrients of experimental soils included 5.0 g kg^−1^ organic matter, 0.6 g kg^−1^ total N, 16.3 mg kg^−1^ available P, and 72.0 mg kg^−1^ available K. The area has a temperate continental monsoon climate, with an annual average temperature of 11.2 °C and precipitation of 779 mm.

A maize cultivar, Luyu 16, and a wheat cultivar, Yanyou 361, were used as crop materials, and maize plants were further studied. The two crops grew in a winter wheat-summer maize rotation system. Winter wheat was directly seeded in October and harvested in June of the next year, and summer maize was directly seeded in June and harvested in October. Four fertilizer treatments were set, that is, CK (no fertilizer), SR (straws from wheat and maize), N (N fertilizer), and SRN (SR plus N fertilizer) (Appendix A). In this experiment, the four treatments and three replicates were assigned in a randomized block design, each plot being 33 m^2^. 

### 4.2. Measurement of Photosynthetic Parameters and Maize Yield

The investigated photosynthetic parameters included net photosynthetic rate, transpiration rate, total conductance to water, intercellular CO_2_ concentration, and total conductance to CO_2_, which were measured using a portable photosynthetic system (LI6400, Li-Cor Inc., Lincoln, NE, USA). Maize ear leaves grew in light, were consistent in appearance, and were selected for parameter determination. The work was done on sunny days from 9:00 to 11:30. Five plants of each treatment were measured, and the average value was taken. Maize yield was estimated according to two rows of 30 individuals of each plot, and the grain yield was standardized to 14% moisture. 

### 4.3. Sampling and Assay of Physiological Indexes

At the maize grain filling stage, the upper parts of ear leaves were collected from three plants of each plot, immediately frozen in liquid nitrogen, and then stored at −80 °C. The samples were used for Illumina sequencing and assay of physiological indexes. Total chlorophyll and soluble sugar contents were measured as described by Yamashita et al. [36] and Wu et al. [37], respectively. Soluble protein content and malondialdehyde (MDA) content were determined with the Bradford assay and thiobarbituric acid method [38], respectively. Activity assays of peroxidase (POD) and catalase (CAT) were performed as described by Hui et al. [39].

### 4.4. RNA Extraction, cDNA Library Construction, Illumina Sequencing, Read Processing, Assembling and Gene Functional Annotation 

RNA Extraction, cDNA library construction, and Illumina sequencing were conducted by WHbioacme Co., Ltd. (Wuhan, China). The procedures for RNA extraction, cDNA library construction, Illumina sequencing, read processing, assembling, and gene functional annotation were the same as described in Liu et al. [14]. 

### 4.5. Real-Time Quantitative RT-PCR 

To validate the reliability of RNA-Seq results, 15 genes were randomly selected, and specific primer pairs were designed (Appendix A). The RNA samples for qRT-PCR analysis were the same as used in the RNA-seq experiments. cDNA was synthesized from 1 μg of total RNA using the Primer Script RT Reagent Kit (Takara, Beijing, China). The total reaction volume for each qRT-PCR was 20 μL. The procedure was 95 °C for 12 min, 40 cycles of 95 °C for 5 s, and finally 60 °C for 32 s. The *ZmActin1*gene was used as a reference, and the relative expression level was calculated using the 2^−ΔΔCt^ method [40].

### 4.6. Statistical Analysis

Unless indicated, all experiments were performed in three independent biological replicates. All data were analyzed by the Excel 2013 software and SPSS.18.0 software (SPSS Inc., Chicago, IL, USA). Statistically, the difference at *p*-value < 0.05 was considered as significant.

## 5. Conclusions

This study provided the transcriptome information of maize ear leaves under long-term SRN tilling mode. As summarized in Figure 5, maize reached its maximum grain yield under SRN, which was due to a series of genes involved in the transport and assimilation of N, P, and K, as well as the mechanisms of sugar, lipid, and biosynthesis of amino acid were regulated in SRN tilling mode and then reduced plants’ nutrient deficiency stresses and increased the photosynthesis of ear leaves at maize grain filling stage. The identification of these promising genes will deepen our global understanding of complex molecular mechanisms among organic waste, mineral nutrient utilization, and crop yield. 

## Figures and Tables

**Figure 1 plants-12-03868-f001:**
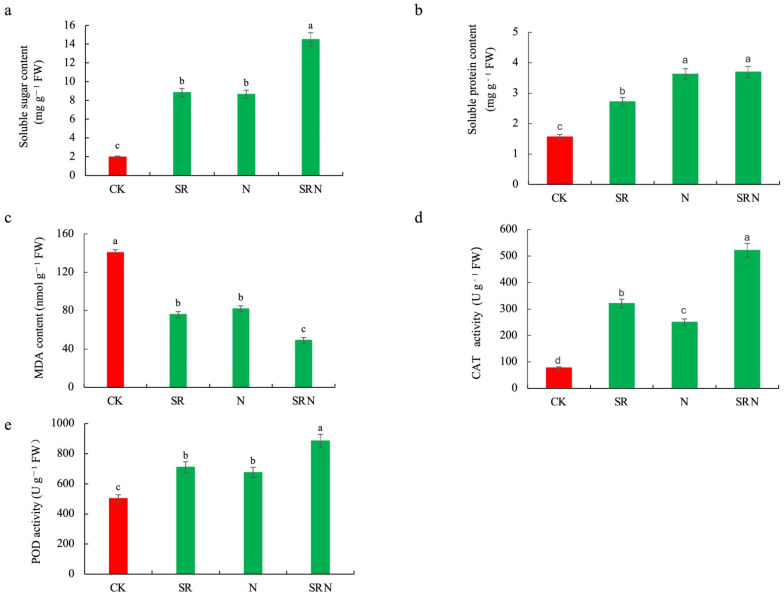
Metabolites and antioxidant enzyme activities analyses responsive to various fertilizer treatments. (**a**) soluble sugar content; (**b**) soluble protein content; (**c**) malondialdehyde content (MDA); (**d**) catalase (CAT); (**e**) peroxidase (POD). Data shown are means ± SE, *n* = 3. Different letters indicate statistically significant differences between treatments at *p*-value < 0.05.

**Figure 2 plants-12-03868-f002:**
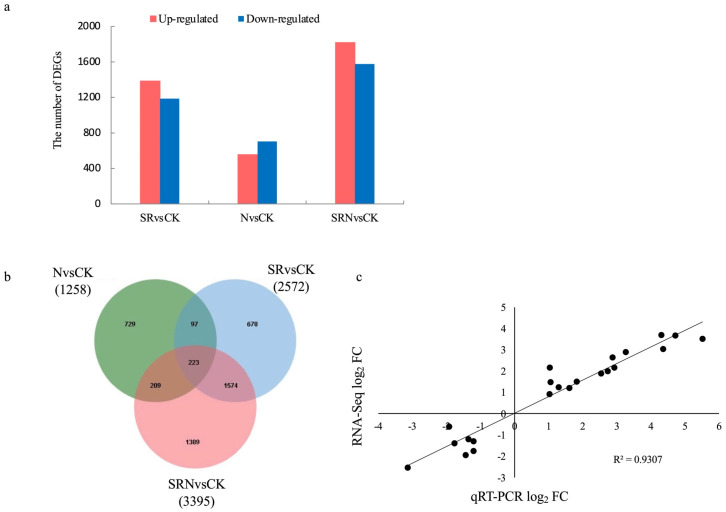
Number of differentially expressed genes (DEGs) from SRvsCK, NvsCK, and SRNvsCK comparisons. (**a**) Distribution of up-regulated and down-regulated DEGs in each comparison; (**b**) Venn diagram analysis; and (**c**) validation of RNA-Seq results.

**Figure 3 plants-12-03868-f003:**
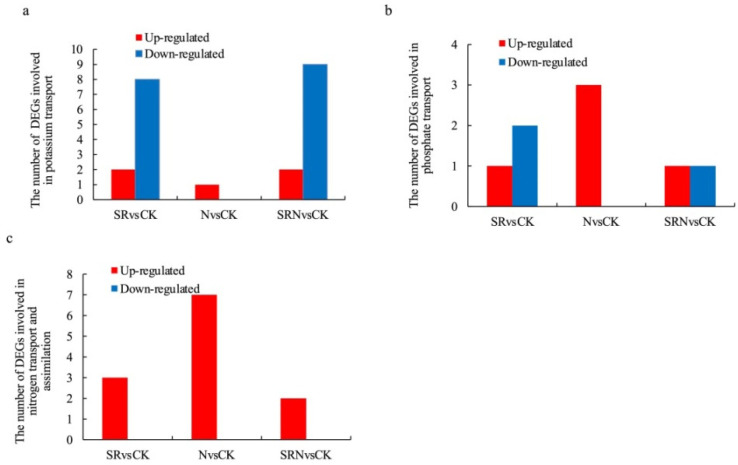
Number of up-regulated and down-regulated DEGs in potassium transport (**a**), phosphate transport (**b**), nitrogen transport, and assimilation (**c**) detected from SRvsCK, NvsCK, and SRNvsCK comparisons (|Log_2_FC| ≥ 1.0, *p*-value < 0.05).

**Figure 4 plants-12-03868-f004:**
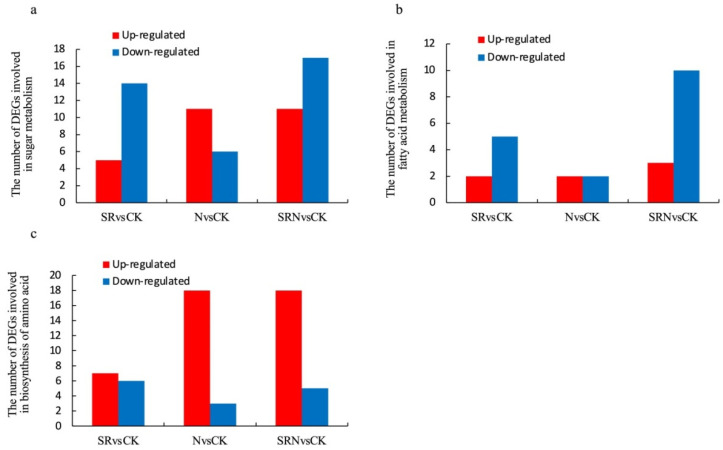
Number of up-regulated and down-regulated DEGs in sugar metabolism (**a**), biosynthesis of an amino acid (**b**), and fatty acid metabolism (**c**) among SRvsCK, NvsCK, and SRNvsCK comparisons (|Log_2_FC| ≥ 1.0, *p*-value < 0.05).

**Figure 5 plants-12-03868-f005:**
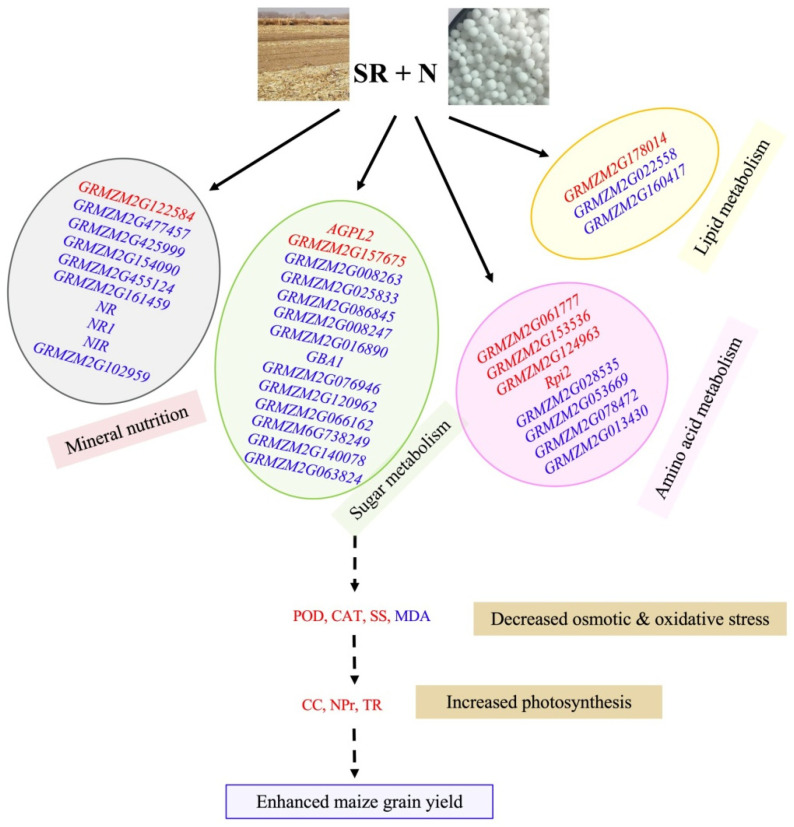
Schematic representation of the molecular mechanism of organic waste, mineral nutrient utilization, and crop yield under SRN tilling mode. Red or blue fonts refer to the up- or down-regulated DEGs in SRN treatment, respectively, over the N treatment. Solid or dashed arrows indicate the direct or indirect effects exerted by SRN treatment, respectively. SR, N, POD, CAT, MDA, SS, CC, NPr, and TR are the abbreviations of straw return, nitrogen, peroxidase, catalase, malondialdehyde, soluble sugar, chlorophyll content, net photosynthetic rate, transpiration rate, respectively.

**Table 1 plants-12-03868-t001:** The years 2019 and 2020 grain yield of maize under long-term fertilizer treatments (kg ha^−1^). Data represent mean ± SE, *n* = 3. Different letters indicate statistically signifcant diferences between treatments at *p*-value < 0.05.

Year	CK	SR	N	SRN
2019	3087 ± 24.5 ^d^	3991 ± 141.1 ^c^	5276 ± 50.7 ^b^	7926 ± 118.4 ^a^
2020	3542 ± 65.6 ^d^	3942 ± 169.7 ^c^	5615 ± 67.9 ^b^	8180 ± 167.6 ^a^

**Table 2 plants-12-03868-t002:** Summary of the DEGs involved in the transport and assimilation of potassium, phosphate, and nitrogen in SRvsCK, NvsCK, and SRNvsCK comparisons (|Log_2_FC| ≥ 2.0, *p*-value < 0.05).

Gene Name	Gene ID	|Log_2_FC| ≥ 2			Annotation
		SRvsCK	NvsCK	SRNvsCK	
Potassium transport					
*GRMZM2G477457*	103635197	−2.29		−2.23	HAK3
*GRMZM2G425999*	103653279	−2.00		−2.36	HAK4
*GRMZM5G838773*	103636639	−2.12			Potassium channel KAT3
*GRMZM2G122584*	103651360	2.36		2.82	Potassium/sodium hyperpolarization-activated cyclic nucleotide-gated channel 1
Phosphate transport					
*GRMZM2G154090*	732716	1.06	2.54	1.03	Pht2
Nitrogen transport and assimilation					
*GRMZM2G455124*	103636218	1.83	2.88	1.61	High-affinity nitrate transporter 2.3
*GRMZM2G161459*	103637664		2.05		NRT1/PTR family 6.3
*NR*	100383210	4.42	8.13	4.31	Nitrate reductase
*NR1*	542278		2.96		Nitrate reductase-like
*NR3*	109939975	2.83			Nitrate reductase 3
*NIR*	542264		2.7		Nitrite reductase 2
*GRMZM2G102959*	103627781		3.89		Ferredoxin—nitrite reductase

**Table 3 plants-12-03868-t003:** Summary of the DEGs involved in sugar, amino acid, and fatty acid metabolism in SRvsCK, NvsCK, and SRNvsCK comparisons (|Log_2_FC| ≥ 2.0, *p*-value < 0.05).

Gene Name	Gene ID	|Log_2_FC| ≥ 2			Annotation
		SRvsCK	NvsCK	SRNvsCK	
Sugar metabolism					
*AGPL2*	103635182	2.93	2.4	2.74	Glucose-1-phosphate adenylyltransferase large subunit 2
*GRMZM2G008263*	100125638	−2.08		−1.47	Granule-bound starch synthase II, a precursor
*GRMZM2G025833*	100281141			−3.13	Beta-amylase
*GRMZM2G086845*	542107	−1.68		−2.29	Fructokinase 1
*GRMZM2G008247*	732807			−2.58	Beta-D-glucosidase precursor
*GRMZM2G016890*	542414		2.68		Beta glucosidase 1
*GBA1*	103634742		2.85		Beta-glucosidase 1
*GRMZM2G076946*	100174972			−4.84	Dhurrinase-like B-glucosidase
*GRMZM2G120962*	103650330			−2.51	4-hydroxy-7-methoxy-3-oxo-3,4-dihydro-2H-1,4-benzoxazin-2-yl glucoside beta-D-glucosidase 1
*GRMZM2G066162*	103641803		3.41		Endoglucanase 12-like
*GRMZM6G738249*	100273093	−1.43		−2.53	Trehalose-phosphate phosphatase J
*GRMZM2G140078*	107546758			−2	Trehalose-6-phosphate phosphatase 2
*GRMZM2G157675*	100282631		−2.08		SWEET
*GRMZM2G063824*	100285023		4.9		Carbohydrate transporter/sugar porter
Fatty acid metabolism					
*GRMZM2G178014*	100191431	1.21		2.37	Alpha/beta-Hydrolases superfamily protein
*GRMZM2G022558*	542018	−3.7		−4.54	Fatty acid elongase 1
*GRMZM2G160417*	100382638			−2.07	Uncharacterized
Biosynthesis of amino acid					
*GRMZM2G028535*	100280719	1.93	3.16	2.82	Delta 1-pyrroline-5-carboxylate synthetase 1
*GRMZM2G061777*	103630430	1.89	1.5	2.08	Delta-1-pyrroline-5-carboxylate synthase-like
*GRMZM2G153536*	100281255		−2.57		Branched-chain-amino-acid aminotransferase
*GRMZM2G124963*	100279149	2.14		2.34	alanine aminotransferase 10
*GRMZM2G053669*	100192350		4.99		Asparagine synthetase 3
*GRMZM2G078472*	100192351		5.52		Asparagine synthetase 4
*Rpi2*	103626247	4.36	3.27	4.72	Ribose-5-phosphate isomerase 2
*GRMZM2G013430*	541959		2.22		Serine acetyltransferase 2

## Data Availability

The datasets generated during and/or analyzed during the current study are available from the corresponding author upon reasonable request.

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
