# Peer review of "Transcriptome Analysis of Maize Ear Leaves Treated with Long-Term Straw Return plus Nitrogen Fertilizer under the Wheat–Maize Rotation System"

_plants, 2023, doi:10.3390/plants12223868_

Round 1
Reviewer 1 Report
Comments and Suggestions for Authors
The authors should provide an explanation on the following observation. While the fertilization processes studied here are meant to provide mineral nutrients to plants, especially NPK, it turns out that this study shows a down-regulation of genes involved in K transport, while it shows up-regulation of Pi transporters as well as nitrate transporters.
Secondly, it would be nice to have a plausible explanation to explain why different gene involved in starch biosynthesis respond differently to the fertilization treatments, in a context of improved photosynthesis.
Thirdly, the representation in figure 5 is only a model and this should be stated. In addition, the 4 arrows on the top may not be equivalent to the 3 in the lower part of the figure: fertilization may more or less directly affect gene expression (despite the fact there may be a cascade of signaling) while the LINK between increased mineral nutrition, improved sugar, aminoacid and lipid metabolisms on one hand, and on the other hand the decrease in oxidative stress up to increased photosynthesis may be far more complex. Actually, the present study does not document the latter phenomena. Therefore, I suggest that the 3 lower arrows are graphically distinguished from the 4 top ones, simply to distinguish what it is reasonably documented here and what is a model. With explanation in the legend.
Minor points:
Line 149, add ‘under’ before ‘all treatments’.
Line 235, replace ‘endeavour’ by ‘more work’.
Line 237, remove ‘the’ before H2PO4
Line 241, ‘this suggests’
Line 247, replace ‘contain’ by ‘are’.
Line 282, ‘… cellulose, which is intriguing…’
Line 303, add ‘relevant’ before DEG
Comments on the Quality of English LanguageSome improvement is necessary
Reviewer 2 Report
Comments and Suggestions for Authors
The manuscript submitted by Li et al. with a title of “Transcriptome Analysis of Maize Ear Leaves Treated with Long-term Straw Return plus Nitrogen Fertilizer under Wheat-Maize Rotation System” represents a significant progress in maize field physiology and transcriptomic analysis. The paper was designed properly and the data were analyzed well with statistical support. The RT-PCR data were further provided support for DEGs identified using RNA-seq data, which makes the work stronger.
I like to point out that it may not be necessary for current paper, however, may be worth for considering future analysis. It is widely known that genes in plants are subject to alternative splicing. It is estimated that >50% of maize genes were alternatively spliced. It is valuable to identify alternatively spliced genes in the current dataset and to examine any differences in different treatments.
Reviewer 3 Report
Comments and Suggestions for Authors
Comments and suggestions are included in the PDF text here attached

Comments on the Quality of English LanguageEnglish language is understandable and easy to read, limited variations/modifications in the formatting are included in the text here attached as pdf file.
Round 2
Reviewer 3 Report
Comments and Suggestions for Authors
The authors have introduced all the requested modifications, except that of the soil composition at the beginning of the experiment responding to this criticism with an acceptable justification.